

# Early maternal separation is not associated with changes in telomere length in domestic kittens (*Felis catus*)

Mikel Delgado[1], C.A. Tony Buffington[1], Melissa Bain[1], Dana L. Smith[2] and Karen Vernau[3]

[1] Department of Medicine and Epidemiology, School of Veterinary Medicine, University of California, Davis, Davis, CA, United States of America
[2] Department of Biochemistry and Biophysics, University of California, San Francisco, San Francisco, CA, United States of America
[3] Department of Surgical and Radiological Sciences, School of Veterinary Medicine, University of California, Davis, Davis, CA, United States of America

## ABSTRACT

**Objective**. Studies of multiple species have found that adverse early life experiences, including childhood trauma and maternal separation, can result in accelerated telomere shortening. The objective of this study was to determine if premature separation from the mother affected telomere length in domestic kittens (*Felis catus*). Subjects were 42 orphaned kittens and 10 mother-reared kittens from local animal rescue groups and shelters. DNA was extracted from whole blood collected from kittens at approximately 1 week and 2 months of age. Telomere length was assessed by qPCR (quantitative polymerase chain reaction) from a total of 86 samples and expressed as a ratio of telomere PCR relative to a single copy gene PCR (T/S).

**Results**. A generalized linear mixed model found there were no detectable differences in telomere length based on survival ($F_{1, 76.2} = 3.35$, $p = 0.07$), orphan status ($F_{1, 56.5} = 0.44$, $p = 0.51$), time point ($F_{1, 43.5} = 0.19$, $p = 0.67$), or the interaction between orphan status and time ($F_{1, 43.5} = 0.86$, $p = 0.36$). Although in other species telomere shortening is commonly associated with aging, even early in life, we did not find evidence for telomere shortening by two months of age. Our results suggest that the experience of early maternal separation in domestic cats who are subsequently hand-reared by humans does not accelerate telomere shortening compared to mother-reared kittens, at least in the first few months of life.

## INTRODUCTION

Early life experiences can impact mental and physical health across the lifespan (*Felitti et al., 1998*). In particular, adverse childhood experiences (ACEs) have received significant attention in the medical and psychological literature because of the potential harm they can cause, including increased risk for future alcohol and drug abuse, depression, and poor health outcomes (*Felitti et al., 1998*). In humans and rodents, inadequate maternal care leads to increased fearfulness and negative affect (*Hane & Fox, 2006*), stereotypies (*Latham & Mason, 2008*), increased fear and panic in adults (*Caldji et al., 1998*; *Nishi, Horii-Hayashi*

Corresponding author
Mikel Delgado,
mmdelgado@ucdavis.edu

& Sasagawa, 2014), and decreased cognitive abilities, including spatial navigation (*Liu et al., 2000*) and memory (*Zalosnik et al., 2014*).

Studies in multiple species have found that adverse early life experiences and other stressors can also result in accelerated telomere shortening (*Chatelain, Drobniak & Szulkin, 2020*; *Cram et al., 2017*; *Gil et al., 2019*; *Nettle et al., 2017*; *Price et al., 2013*; *Xavier et al., 2018*). Telomeres are highly conserved, non-coding regions of repetitive nucleotide sequences at each end of a chromosome that protect the genome during cell replication by preventing genes from truncating. Cells have a limited reproductive capacity, referred to as the "Hayflick Limit" (*Hayflick & Moorhead, 1961*). Telomeres shorten during mitosis and once they reach a threshold the cell stops reproducing and enters a state of senescence or dies (*Allsopp et al., 1995*). Because of this process, telomeres also are associated with aging and the remaining capacity of a cell to replicate (*Aubert & Lansdorp, 2008*; *Chiu & Harley, 1997*). Telomeres are subject to more shortening early in life (*Frenck, Blackburn & Shannon, 1998*), and telomere length is a strong predictor of lifespan and mortality in multiple species (*Bize et al., 2009*; *Cram et al., 2017*; *Heidinger et al., 2012*). Thus, adverse experiences early in life may have more influence on telomere length and long-term health and life-history than do adverse experiences later in life (*Ridout et al., 2017*).

Previous studies have demonstrated the effects of adverse early life experiences on telomere length. Among European starlings, an early life competitive disadvantage (being smaller than other brood members) led to more rapid telomere attrition by 12 days of age compared to birds who were larger than the rest of their brood (*Nettle et al., 2015*). Meerkat pups born into larger groups of young have shorter telomeres due to increased competition for access to milk (*Cram et al., 2017*). Maternal stress also can shorten telomeres. For example, an increased rate of telomere loss was found in female offspring of laying zebra finches treated with corticosterone compared to controls and the offspring of zebra finches treated with 17-$\beta$-estradiol (*Tissier, Williams & Criscuolo, 2014*).

Social isolation shortens telomeres (*Aydinonat et al., 2014*), but there have been few studies of the effects of maternal presence or care on telomere length. Children in rural areas of China who were cared for by extended family had shorter telomeres compared to children raised with their mothers (*Chen et al., 2019*), and individuals who described their parents as neglectful had shorter telomeres than those who did not (*Enokido et al., 2014*). Similar effects are found in non-humans; Welsh pony foals that were abruptly weaned and separated from their mothers showed higher levels of stress and shorter telomeres compared to foals who were progressively weaned (*Lansade et al., 2018*). Rhesus macaques reared by their mothers had longer telomeres than monkeys who were raised with peers or in isolation (*Schneper et al., 2016*).

Kittens have frequent interactions and physical contact with their mother during the nursing and weaning periods, which typically last until at least two months of age (*Albonetti, 1988*). Some kittens become inadvertently separated from their mothers before weaning and are subsequently hand-reared by humans. Hand-raising of kittens has become increasingly common as animal shelters improve their ability to care for orphaned neonatal kittens. Orphaned neonatal kittens have a high rate of mortality (*Little, 2013*), are at risk for abnormal behaviors (*Delgado, Walcher & Buffington, 2020*), and show increased distress

vocalizations and activity during a nest separation (*Lowell et al., 2020*), compared to mother-reared kittens. To better understand the effects of early maternal separation on domestic cats, the purpose of the described study was to investigate whether premature separation from the mother affected telomere length in kittens at two time points.

Previous studies in adult cats have established a relationship between telomere length and aging (*McKevitt et al., 2003*), and disease processes such as kidney failure (*Quimby et al., 2013*). These and other studies have established the feasibility of measuring telomere length in cats (*Brümmendorf et al., 2002*; *Pang & Argyle, 2009*). To our knowledge, this is the first study investigating the effect of early maternal separation on telomere length and telomere dynamics in kittens.

If maternal separation has deleterious effects, orphaned kittens could show shorter telomeres compared to control kittens. We expected similar telomere length in both groups in the first week of life, as all kittens who were separated from their mothers would have experienced the separation recently. We predicted that all kittens would experience some telomere shortening by eight weeks of age due to the normal process of aging, and we specifically hypothesized an interaction effect, such that kittens who were separated from their mothers early in life (orphans) would have significantly shorter telomeres than kittens raised by mothers (mother-reared) at eight weeks of age.

## Methods

All animal procedures were approved by the Animal Care and Use Committee at the University of California, Davis, under Protocol #20379.

## Study animals

Subjects were 42 orphaned kittens (OR) and 10 mother-reared (MR) kittens from local rescue groups and shelters. Orphans had been turned into shelters or rescue groups without their mothers by members of the public. All kittens were entered into the study before they were 7 days of age. Age was assessed by an experienced veterinary technician or shelter rescue staff, as determined by known date of birth, or as estimated by presence of umbilical cord, weight, ear position, and degree of eyes opening (*Little, 2011*). Orphaned kittens were raised in foster homes and were housed in incubators set to 80–90 °F, and with relative humidity set to 50–60% (*Little, 2013*; *Peterson, 2011*). Incubators included a heating pad and bedding. All orphans were fed commercial kitten formula (Breeder's Edge®, Revival Animal Health, Orange City, Iowa) until weaned, when they were fed a commercial dry and wet food (Purina Pro-Plan Kitten Formula®, St. Louis, MO). Mother-reared kittens were also cared for in foster homes; they consumed their mother's milk until weaned onto the same type of wet and dry kitten food as the orphans received. Foster caretakers gave written consent for fostered kittens to participate in the study. Nine OR kittens died while in foster (average age of death: 20 days, SD: 11.9 days, range 6 to 42 days); the remaining kittens were adopted into homes after being weaned and neutered.

## Experimental procedures

Blood (200 μl) was drawn from each kitten via jugular venipuncture by a veterinarian or veterinary technician when kittens were approximately 1 week and two months of age

**Table 1** **Number of blood samples acquired and used in analyses from mother-reared and orphaned kittens at Time 1 (one week) and Time 2 (two months).** The number of repeated samples from each group is indicated in lighter text and parentheses.

| Source | Samples not used in final analyses | Analyzed samples Time 1 | Analyzed samples Time 2 | Total samples collected |
|---|---|---|---|---|
| Mother-reared kittens | 3 lost due to experimenter error | 6[a] | 10[b] | 19 |
| Repeated samples | | (6) | (6) | |
| Orphaned kittens | 4 assay development 3 insufficient DNA | 37[c] | 33[d] | 77 |
| Repeated samples | | (28) | (28) | |
| Total | 10 | 43 | 43 | 96 |

**Notes.**

Note: all ten samples not used in final analyses were acquired from Time 1.
[a] samples from 3 litters.
[b] 4 litters.
[c] 16 litters.
[d] 17 litters.

(mean age of first blood sample: OR: 8 days, SD: 2 days, MR: 10 days, SD: 1.2 days, range 6–13 days; mean age of second blood sample: OR: 69 days, SD: 9.3 days, MR: 59 days, SD: 4.3 days, range 50–93 days). Blood samples were placed into EDTA tubes then transferred to a cryotube and stored at −80 °C until analysis.

A total of 96 blood samples were collected, and 86 samples were used in the final analysis. Blood from 28 OR and six MR kittens was collected and analyzed at both time points. Only one sample (from Time 1) was collected and analyzed from each of the nine orphans who died. There were an additional five OR and four MR kittens from whom blood was collected and analyzed only for Time 2. Three samples (all orphans) did not produce sufficient DNA, and four samples were used during assay development and were not included in the final dataset. Due to experimenter error, three of the blood samples from MR kittens were misplaced, which left 86 viable samples for analysis (Table 1).

## Quantification of telomere length

The telomere length measurement assay was adapted from Cawthon (*Cawthon, 2002*; *Park et al., 2013*). Telomere lengths were assessed by qPCR (quantitative polymerase chain reaction) and expressed as a ratio of telomere PCR relative to a single copy gene PCR (T/S). The qPCR analysis was done on a Roche 480 LightCycler qPCR machine, using the 'Absolute quantification/2nd derivative max' automated algorithm, which calculates the point at which the maximum change in slope of the fluorescence amplification curve becomes the crossing point. The 'T' qPCR reaction and 'S' qPCR reactions were carried out sequentially.

The measurements were made from DNA extracted from whole blood. DNA extract samples were stored at −80 °C. The DNA extraction includes an RNAse step and was carried out in a biosafety cabinet. The quality of the DNA was measured spectroscopically, using a Nanodrop2000. Genome integrity for these samples was verified by separating 100ng of each DNA extract on a .8% agarose gel and no degradation was observed. All

samples used for telomere measurement met the criteria, 260/280 nm ratios, $1.7 < n < 2.0$, 260/230 nm ratios $>1.0$, and were considered free of contaminants.

Whole blood was treated with a 'cell lysis' solution (Qiagen, QIAmp DNA blood mini kit, #51106) and the lysate was spun through a column that retained the DNA. Subsequent washing and elution released the purified DNA from the column. The DNA extracted from blood from each animal was diluted to 10 ng/ul in a 96-well stock plate. A liquid handler was used to pipet triplicate reactions from the 96-well DNA plate to a 384-well 'T' (telomere) reaction plate, each well containing 7.5 ul of PCR reaction mixture to give 2.5 ng/ul final DNA concentration. Immediately after the T reactions were put into the PCR machine, the same DNA stocks were used to pipet triplicate reactions on a separate 384-well 'S' (single copy gene) reaction plate.

For the feline single copy gene (S), primers were made to amplify feline GAPDH. For the forward primer, we used the sequence (5′–>3′) GTGGTGAAGCAGGCATCAGA and for the reverse primer, we used CACTGTTAAAGTCGCAGGAGACA. Both primers were used at a concentration of 1 μM in the PCR reaction. The primers used for the telomere (T) PCR were Tel C (5′-TGTTAGGTATCCCTATCCCTATC-3′), at a final concentration of 200 nM, and Tel G (5′-ACACTAAGGTTTGGGTTTGGGTT-3′), at a final concentration of 400 nM.

Both telomere and single copy gene PCR reactions were performed with QuantiFast SYBR Green PCR Kit (QIAGEN) on a Roche LightCycler 480 Realtime PCR machine (LC480; Roche Diagnostics Corporation, Indianapolis, IN). T and S reactions were 10ul with stock 2x Quantifast mix (5 ul/reaction). The Quantifast mix contains a proprietary fluorescent polymerase for PCR. The concentration of DNA in all experimental reactions was 2.5 ng/ul.

The thermal PCR cycling profiles were as follows: for T (telomeric), denature at 95 °C for 15 min, one cycle; denature at 94 °C for 15 s, anneal/extend at 49 °C for 15 s, one cycle. Denature at 94 °C for 15 s, anneal/extend at 49 °C for 30 s, one cycle. Melt at 95 °C for 15 s, anneal at 62 °C for 10 s, 74 °C for 15 s with fluorescence data collection 32 cycles. For S (single copy gene), denature at 95 °C for 10 min, one cycle; denature at 94 °C for 15 s, anneal at 60 °C for 60 s with fluorescence data collection, 45 cycles. A standard curve was created from two-fold serial dilutions of pooled genomic DNA from four OR kittens, from which the concentrations of telomere reactions and single copy gene reactions were determined. A 5-point, 2x-dilution standard curve was made of pooled genomic DNA from four OR kittens, starting with 10ng/ul, and ending with .312ng/ul. At least three runs (T and S paired runs) were carried out for each sample. The efficiency of every T run was greater than 90% and the efficiency of every S run was greater than 95%. The $R^2$ For T1, S1, T2, and S2 runs were 1.0, 0.99, 0.99, and 1.0 respectively.

Genomic DNAs from the same 4 kittens were plated individually as positive controls, to serve as quality controls for each run. Three pairs of T and S PCR reactions were run on different days and the T/S PCR ratio was calculated for each pair. If, after three runs, a value looked like an outlier, it was discarded and an additional run was made. For this plate of samples, the inter-plate CV (the standard deviation/average of the 2 T/S values) for every sample was averaged and came to 3%.

## Statistical analyses

All data were analyzed using SAS University Edition (SAS Institute Inc, Cary, NC). To assess the effect of orphan status and age on telomere length, we used a linear mixed model (LMM) in PROC MIXED including survival (Y/N), orphan status (Y/N), time point (1 or 2), and the interaction between orphan status and time point as predictors, with telomere ratio as the dependent variable. The residuals of the LMM met the assumptions of normality (q-q plot) and homogeneity (Levene's test, $p > 0.05$), and data from all groups also met the assumption of equality of variance (all Satterthwaite tests, $p > 0.05$). Litter and kitten ID (nested within litter) were included as random effects to account for repeated measures of the same individuals and the effects of relatedness on kittens from the same litter. Telomere lengths followed a normal distribution (Kolmogorov–Smirnov test $= 0.07$, $p > 0.15$) and we used a Pearson's correlation to assess the relationship between telomere lengths at time points 1 and 2 for kittens who had been sampled twice. We also performed a post-hoc sensitivity analysis in G*Power 3.1.9.2 (University of Kiel, Germany; *Faul et al., 2007*) to determine the effect size that would have been necessary to find a difference between groups. Figures were created in R 4.0.2 (R Foundation for Statistical Computing, Vienna, Austria).

## RESULTS

Telomere ratios ranged from 0.63 to 1.44 ($x = 0.95$, $SD = 0.14$). The model found no effect of survival ($F_{1,76.2} = 3.35$, $p = 0.07$), orphan status ($F_{1,56.5} = 0.44$, $p = 0.51$), time point ($F_{1,43.5} = 0.19$, $p = 0.67$), or the interaction between orphan status and time point ($F_{1,43.5} = 0.86$, $p = 0.36$; Fig. 1). The results were the same when assessing only the 34 kittens from whom we had two blood samples (orphan status ($F_{1,45.4} = 0.75$, $p = 0.39$), time point ($F_{1,32} = 0.01$, $p = 0.91$), the interaction between orphan status and time point ($F_{1,32} = 0.72$, $p = 0.40$)).

There was a small but statistically significant correlation between telomere length at Time 1 and Time 2, $r$ (34) $= 0.36$, $p = 0.04$; Fig. 2. When comparing by orphan status, the correlation for MR kittens was $r$ (6) $= 0.80$ ($p = 0.06$), and for OR kittens was $r$ (28) $= 0.28$ ($p = 0.14$), neither of which was statistically significant.

## Sensitivity analysis

Because of the null findings between groups, the post-hoc assessment of the sensitivity of the analysis allowed us to determine the effect size that would have been necessary to find a difference between groups. With adequate power ($\beta = 0.80$, $\alpha = 0.05$), to find a statistically significant difference between the number of OR and MR kittens we examined at each time point would have required a large effect size (Cohen's $d = 1.11$ and 0.88 respectively). This would have equaled a difference in T/S ratio between groups of at least 0.13 for Time 1 and 0.14 for Time 2. For the pooled data for all kittens, to find a difference between the two time points would have required a medium effect size of Cohen's $d = 0.54$, which would have equaled a difference between groups in T/S ratio of at least 0.08. The means of both groups were very close (0.98 and 0.97 for Time 1; 0.98 and 0.93 in Time 2), and the differences resulted in effects sizes of 0.07 (Time 1) and 0.35 (Time 2). If our findings came
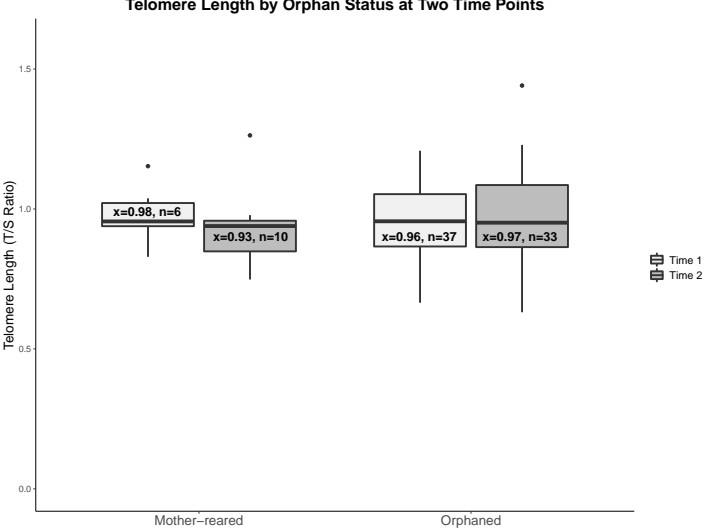

**Figure 1** **Boxplots of telomere lengths with sample sizes for mother-reared and orphaned kittens at 1 and 8 weeks of age summarized from raw data.** Boxplots of telomere lengths with means and sample sizes for mother-reared and orphaned kittens at 1 and 8 weeks of age summarized from raw data. Boxplots include median value, IQR (interquartile range), and outliers.

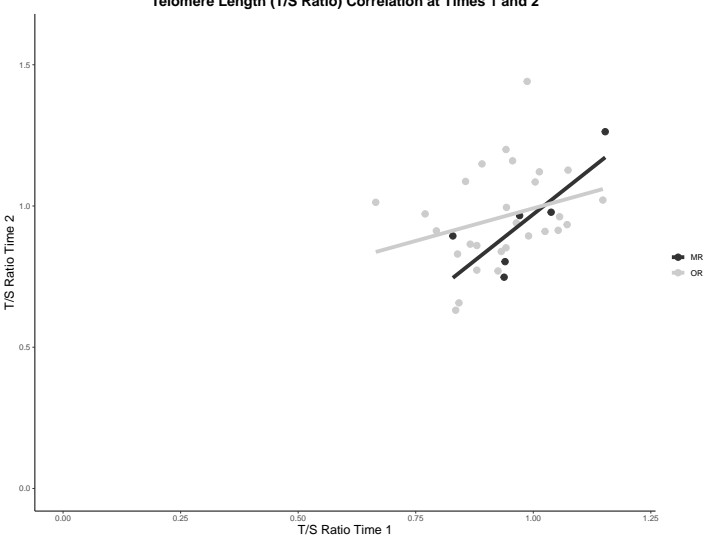

**Figure 2** **Scatter plot of correlations between telomere lengths at two time points.** Pearson's correlations between telomere lengths at approximately one week and two months of age for six mother-reared kittens ($r(6) = 0.80$, $p = 0.06$) and 28 orphaned kittens ($r(28) = 0.28$, $p = 0.14$) for whom repeated samples were collected. The overall correlation for all repeated samples was $r(34) = 0.36$, $p = 0.04$.

from representative samples, we would have needed samples from more than 5000 and 200 kittens, respectively, to identify a statistically significant effect at either time point.

## DISCUSSION

We found no detectable differences in telomere length between orphaned and mother-reared kittens. We predicted that by approximately two months, orphaned kittens would have shorter telomeres than mother-reared kittens and would experience more telomere attrition. Although telomere shortening is commonly associated with aging, we also did not find evidence for telomere shortening in either group of kittens between the first week of life and approximately two months later.

Some previous studies have found deleterious effects of early maternal separation, including telomere shortening, in other species (*Chen et al., 2019*; *Enokido et al., 2014*; *Lansade et al., 2018*; *Schneper et al., 2016*). Previous studies have found that being orphaned increases behavioral signs of stress and other abnormal behaviors compared to mother-reared kittens (*Delgado, Walcher & Buffington, 2020*; *Lowell et al., 2020*). However, for domestic cats, being raised by humans in the absence of the mother may also have some protective effects. Kittens are altricial and unlikely to survive when orphaned without outside care. Being raised by humans may be more like progressive weaning; foals who were gradually weaned experienced less telomere shortening than abruptly weaned foals (*Lansade et al., 2018*). All of the orphans in our study were maintained at consistent temperatures and humidity and were bottle-fed, which may have lowered stress and prevented competition for food as is experienced in many young animals (*Hudson & Trillmich, 2008*), possibly preventing telomere loss. Further research is needed to better understand whether human care can compensate for the impact of maternal loss in kittens.

Alternatively, it is possible that the mothers in our study were under an increased stress level, thereby shortening the telomeres of the mother-reared kittens. This explanation seems unlikely as all mother cats were social with humans, and housed in quiet, private, indoor environments. Instead, it appears that any potential stress in cats related to being orphaned does not have a large impact on telomere length. Cortisol reduces the production of telomerase (*Choi, Fauce & Effros, 2008*), an enzyme which may preserve telomere length (*Boccardi & Paolisso, 2014*). Studies of stress in adult cats have not consistently found a strong correlation between cortisol and other signs of stress (*Fukimoto et al., 2020*; *Gourkow et al., 2014*), suggesting one potential explanation for why orphaned kittens may show more behavioral signs of stress but no telomere shortening compared to mother-reared kittens. Further studies are needed to explore whether stress increases cortisol level in neonatal kittens.

A longitudinal approach is useful for assessing the effects of aging on telomere length. We did not track kittens into adulthood; ideally, measurements of survival and telomere length would be taken from cats at multiple time points to assess whether premature maternal separation has any long-term effects on health. Among kittens who survived to two months of age (100% of mothered kittens and 79% of all orphans), we found no evidence of an impact of being orphaned on telomere length. It is possible that our inability to find signs

of telomere shortening in either study group may have been due to the short time interval between sample collection (approximately 8 weeks). Longitudinal studies of young birds, (*Nettle et al., 2017*; *Nettle et al., 2015*) sheep (*Fairlie et al., 2016*), adult laboratory mice (*Cai et al., 2015*), and Sudanian grass rats (*Grosbellet et al., 2015*) found significant shortening of telomeres within similar or smaller time frames (*Bateson, 2016*). However, we cannot know if rates of attrition across species are comparable.

Rather than finding telomere shortening, we found a correlation between relative telomere length at each time point. Kittens with relatively shorter telomeres at Time 1 also had relatively shorter telomeres at Time 2. Some orphans appeared to have increased relative telomere length, whereas a decreased relative telomere length was found in others, suggesting more variability among orphans.

Our study had some limitations. We had difficulty procuring sufficient blood samples from mother-reared kittens, leaving us with imbalanced groups and power sufficient only to detect medium to large effect sizes. Our sample may also be biased by the fact that we could not re-test nine kittens (all orphans) who died after their initial blood sample was collected. Our model suggested that the initial telomere lengths of kittens who died were not different from the rest of the population sampled, and we did not find any differences between surviving orphans and mother-reared kittens. Finally, there are different ways to assess telomere length, including qPCR, Q-FISH, TRF analysis, and TeSLA (*Lai, Wright & Shay, 2018*; *Montpetit et al., 2014*). qPCR offers the benefits of being easy to perform and requiring small amounts of DNA. However, qPCR can only provide relative, rather than absolute telomere length, and is less sensitive to short telomeres or telomeropathies (*Gutierrez-Rodrigues et al., 2014*; *Lai, Wright & Shay, 2018*). Thus, using qPCR may have reduced our ability to find differences between our two populations of interest.

## CONCLUSIONS

Although there may be some reported deleterious effects of early maternal separation in domestic cats (*Ahola, Vapalahti & Lohi, 2017*; *Delgado, Walcher & Buffington, 2020*; *Lowell et al., 2020*), we did not find that orphaned kittens had shorter telomeres than non-orphans at approximately one week or two months of age. If differences do exist between the two groups, they are likely small and of unknown clinical significance. Further research, ideally with larger samples, should assess whether there are differences between orphaned and mother-reared populations later in life.

**Abbreviations**

| | |
|---|---|
| **MR** | mother-reared |
| **OR** | orphaned |
| **PCR** | polymerase chain reaction |
| **qPCR** | quantitative polymerase chain reaction |
| **S** | single copy gene |
| **T** | telomere |

## ACKNOWLEDGEMENTS

We thank everyone who assisted with the completion of this research, including: Dr. Jue Lin from the Blackburn Lab at UCSF; Cara Wademan, Samantha Barnum and Emir Hodzic from the UC Davis Veterinary Genetics Laboratory; Dr. Casey Kohen, Meghan Ramczyk, Dallas Butterfield, Emma Hewitt, Robert Collins, Kathy Pinkston, Marcy Vaughn, and Abrah Wymore. We especially thank the foster caretakers and rescue groups who cared for the kittens in this study. We also thank the three reviewers whose constructive feedback helped to improve the manuscript.

### Funding

This work was funded by a grant from the Koret Shelter Medicine Program/Center for Companion Animal Health at the University of California, Davis. Mikel Delgado was supported by funding from Maddie's Fund and the National Center for Advancing Translational Sciences, National Institutes of Health, through grant number UL1 TR001860 and linked award TL1 TR001861. The funders had no role in study design, data collection and analysis, decision to publish, or preparation of the manuscript.

### Grant Disclosures

The following grant information was disclosed by the authors:
Koret Shelter Medicine Program/Center for Companion Animal Health at the University of California, Davis.
Maddie's Fund and the National Center for Advancing Translational Sciences, National Institutes of Health: UL1 TR001860.

### Competing Interests

The authors declare there are no competing interests.

### Author Contributions

- Mikel Delgado conceived and designed the experiments, performed the experiments, analyzed the data, prepared figures and/or tables, authored or reviewed drafts of the paper, and approved the final draft.
- C.A. Tony Buffington and Melissa Bain conceived and designed the experiments, authored or reviewed drafts of the paper, and approved the final draft.
- Dana L. Smith performed the experiments, authored or reviewed drafts of the paper, and approved the final draft.
- Karen Vernau conceived and designed the experiments, performed the experiments, authored or reviewed drafts of the paper, and approved the final draft.

### Animal Ethics

The following information was supplied relating to ethical approvals (i.e., approving body and any reference numbers):

All animal procedures were approved by the Animal Care and Use Committee at the University of California, Davis (Protocol 20379).

## Data Availability

The datasets used for this study are available from Dryad: Delgado, Mikel; Buffington, C.A.T. Tony; Bain, Melissa; Smith, Dana L.; Vernau, Karen (2020), Early maternal separation is not associated with changes in telomere length in domestic kittens (*Felis catus*), Dryad, Dataset, https://doi.org/10.25338/B8PD0T.

## Supplemental Information

Supplemental information for this article can be found online at http://dx.doi.org/10.7717/peerj.11394#supplemental-information.

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
