# Peer review of "Early maternal separation is not associated with changes in telomere length in domestic kittens (Felis catus)"

_PeerJ, doi:10.7717/peerj.11394_

## Round 0.1 · original submission · Major Revisions

Two reviewers evaluated your paper, and both of them acknowledge the importance of your study. However, they also find some problems with the manuscript and made constructive comments to improve the manuscript. Please revise your manuscript based on their comments. Please bear in mind that appropriate statistical analysis is important to consider for acceptance.

Reviewer 1 ·

Basic reporting

The use of language is for the most part clear and professional.

There is adequate literature cited and sufficient background. I would like to see some references supporting the statements L37-38. It may help the reader if the authors noted that most of the literature in the introduction concerning early life adversity and telomeres is from research in wild animals, while the current MS studies domestic cats. This is an important contrast that could be explored (possibly in the discussion where human rearing of kittens is detailed).

The current study also deals with a particular type of early life adversity: loss of the mother. While to my knowledge there are no studies (wild or domestic) of mother-loss and telomeres, there may be studies of mother loss and other metrics of stress (stress hormones for example) and I would urge the authors to briefly research this and cite any relevant work. This will help build a case that mother-loss specifically (rather than adversity in general) has health consequences. Stress is also known to accelerate telomere loss.

The article structure is good. A section is missing: statistical analysis details in the methods. In my view this is vital for the reader to know the details of the analyses that have been carried out. For example, L105 reports a correlation but there is no info about how this p-value was generated and the reader therefore cannot know if it is legitimate. The authors must add a methods section detailing ALL statistical tests and analyses. In each case, the dataset and sample size, the analysis performed and info about the response and predictor terms and random terms etc.

The figures look good. The y-axis in both figures should be labelled "telomere length (T/S ratio)". The black regression line in fig2 should not extend beyond the bounds of the black datapoints. It would be helpful to see the sample sizes for each box on fig1. The raw data is present and looks straightforward to interpret.

The MS is self-contained and has well-conceived hypotheses and relevant results.

Experimental design

I am satisfied that this is primary research within the journal scope. The research is well defined and fills a knowledge gap.

I am satisfied that ethical standards were high.

As stated above, I am not satisfied that adequate description of the statistical methods is provided, and without the specific statistical info the analyses could not be replicated.

I would say the sample sizes are at the limit for rigorous tests, and this should be openly stated and the implications discussed.

Validity of the findings

I have two concerns about the sample sizes.

1) The number of samples for mother-reared offspring is very small, and quite imbalanced compared to OR kittens. The authors should state this and discuss possible implications in the discussion. The sensitivity analysis is a good addition.

2) The precise breakdown blood sample availability from the two groups is too confusing. L59 states that 10 mother-reared (MR) kittens were used. Line 76 states that 9 MR kittens had blood sampled at both timepoints, but figure 2 only shows 5 MR points, and the datafile appears to show 6 MR kittens sampled at both timepoints. L77 states there were an additional 4 MR kittens with blood samples at only one time point. To me this suggests 13 MR kittens were used, not 10 as reported L59. The data appears to contain 11 unique MR kittens. This confusion leaves the reader unclear about what the sample sizes were for different tests, and gives the impression of somewhat sloppy data handling. It is crucial this is corrected.

I would like to see the analyses repeated excluding the kittens that died, or with a predictor added for "survived to 8weeks Y/N". It is possible these kittens were already ill, biasing the results. It is also possible that their telomeres shortened so much that they died, thus masking the true effect because of the absence of their data at the second timepoint. I would like to see discussion of this and urge the authors to research 'selective disappearance'.

It is unclear whether analyses contain samples from the same kittens at the two time points. As stated above, this must be much clearer. If this is the case, kitten ID must be included as a random term, and should be nested within litter ID as a random term.

Additional comments

Overall I commend the authors on an interesting topic. While it appears adversity does shorten telomeres in wild animals, careful rearing by humans can remove these ill-effects. This is a useful and interesting finding. It's clear a lot of work has been conducted. While I am not an expert on telomere laboratory analysis, this appears to have been done carefully and correctly. There is a weakness in the data handling and analysis and I urge the authors to be careful and transparent to ensure the MS is ready for publication.

·

Basic reporting

This study is investigating the effects of maternal separation on early-life telomere length and telomere dynamics in kittens, which has not been investigated so far. It is therefore filling a gap in the literature. However, a lack of context in the abstract and the introduction, together with a not sufficiently detailed state-of-the-art and a too few number of studies cited, are masking the importance of this study. I provide detailed suggestions along with references (with 10 examples, but the authors are of course free to extend this suggestion and to choose other papers than those I suggested) that should be included to improve the background and context of this study.
In addition, my ability to assess the strength and validity of the findings at that stage is hampered by a lack of various key information in the methods. I provide some suggestions and have several questions that should help the authors advancing this issue. I hope the authors will find my comments helpful. I remain available for further exchanges or review of this article, would it be needed.
With best regards,
Mathilde Tissier


Clear, unambiguous, professional English language used throughout.
Yes

Intro & background to show context.
1. Abstract: although your objective and results are clear, I think that the abstract is lacking context. You should add 1-2 lines of background before your objective and 1-2 lines after your results to replace your study into a broader context, so the reader might understand rapidly the extent of your work. In addition, you should state on which cell you measured telomere length (L21): white blood cells? For the results, it seems to me that your sentences L25-29 are redundant with what you said L24-26. I would keep only the sentence L25-29 that seem more adapted for an abstract. This would provide space for a sentence replacing your results into a broader context.

2. Introduction: I find the introduction well written. Perhaps consider starting with a broader context, not directly and solely on telomeres. For instance, you could broaden your sentence L33-34 by stating that “Studies in multiple species have found that adverse early life experiences can result in modifications of adults’ behavioral, biochemical or molecular phenotypes. For instance, studies have pinpointed that telomeres shortening is increased in individuals facing harsh early life conditions”. In addition, you do not mention the potential relationship nor the existing knowledge/literature on the link between maternal effects, maternal separation and telomere length/telomere dynamics in other species, which is at the core of your study. You should include a paragraph on this (L42-43, between the state-of-the art on early life experiences and your specific hypothesis on the effect of maternal separation on kittens’ telomere length).

Below, I suggest some minor edits regarding the introduction.

Minor comments on the introduction
L35-37- consider revising. Here is a suggestion: Cells have a limited reproductive capacity, referred as the “Hayflick Limit”5. Telomeres shorten during mitosis and once they reach a threshold, the cell stops reproducing and enters a state of senescence or dies 6.

L38-39 – Telomeres are also

L41 – you mention health and well-being but you only cited studies referring to lifespan and mortality. Consider replacing well-being by “life-history” and to include studies connecting telomeres to health to support your statement.

L43-53 – references on other taxa are critically lacking to support your hypothesis and predictions in this section (see my suggestions).

L47 – are subsequently hand-reared by humans

L48-49 – no, you cannot say that this is the first study on telomeres in kittens; see Brümmendorf, T. H., Mak, J., Sabo, K. M., Baerlocher, G. M., Dietz, K., Abkowitz, J. L., & Lansdorp, P. M. (2002). Longitudinal studies of telomere length in feline blood cells: implications for hematopoietic stem cell turnover in vivo. Experimental hematology, 30(10), 1147-1152 > this is just one example.
You should rephrase this sentence, and make it more specific to the exact objective of your study, which is investigating the effects of maternal separation on early-life telomere length and telomere dynamics in kittens, which has not been investigated so far.

L49 – telomere length

L54 – Please include a prediction on the dynamics of telomeres, in addition to your predictions on telomere length (since you mention telomere shortening in your abstract and introduction + results)

Literature well referenced & relevant.
The literature on telomeres is extremely vast and I don’t expect you to cite all the articles published, but your state-of-the art should be a bit more referenced, including key papers, especially for the relationships between telomere length and life-history traits and health, on how telomere shortening in early-life may have long-term consequences or on the relationship between maternal effects and telomeres. You only have 22 references, which is very few, and offers room to cite more papers in your introduction.

Some examples (among which reviews or meta-analyses):

a) Relationship between telomeres and health or life-history traits

Monaghan P, Haussmann MF. 2006 Do telomere dynamics link lifestyle and lifespan? Trends Ecol. Evol. 21, 47–53. (doi:10.1016/j.tree.2005.11.007)

Bize P, Criscuolo F, Metcalfe NB, Nasir L, Monaghan P. 2009 Telomere dynamics rather than age predict life expectancy in the wild. Proc. Biol. Sci. 276, 1679–83. (doi:10.1098/rspb.2008.1817)

Seeker, L. A., Ilska, J. J., Psifidi, A., Wilbourn, R. V., Underwood, S. L., Fairlie, J., ... & Whitelaw, B. (2018). Bovine telomere dynamics and the association between telomere length and productive lifespan. Scientific reports, 8(1), 1-12.


b) Relationship between early life experiences, stress and telomeres

Price, L. H., Kao, H. T., Burgers, D. E., Carpenter, L. L., & Tyrka, A. R. (2013). Telomeres and early-life stress: an overview. Biological psychiatry, 73(1), 15-23.

Chatelain M, Drobniak SM, Szulkin M. 2020 The association between stressors and telomeres in non-human vertebrates: a meta-analysis. Ecol. Lett. 23, 381–398. (doi:10.1111/ele.13426)

Ilska-Warner, Joanna J., et al. "The genetic architecture of bovine telomere length in early-life and association with animal fitness." Frontiers in genetics 10 (2019): 1048.

c) Relationship between maternal effects, maternal separation, telomeres in early life and phenotypes

Haussmann MF, Longenecker AS, Marchetto NM, Juliano SA, Bowden RM (2011) Embryonic exposure to corticosterone modifies the juvenile stress response, oxidative stress and telomere length. Proc R Soc Lond B Biol Sci 279:

Tissier, M. L., Williams, T. D., & Criscuolo, F. (2014). Maternal effects underlie ageing costs of growth in the zebra finch (Taeniopygia guttata). PloS one, 9(5), e97705

Chen, X., Zeng, C., Gong, C., Zhang, L., Wan, Y., Tao, F., & Sun, Y. (2019). Associations between early life parent-child separation and shortened telomere length and psychopathological outcomes during adolescence. Psychoneuroendocrinology, 103, 195-202.

BOTHA, Martmari, GRACE, Laurian, BUGARITH, Kishor, et al. The impact of voluntary exercise on relative telomere length in a rat model of developmental stress. BMC research notes, 2012, vol. 5, no 1, p. 697.

Structure conforms to PeerJ standards, discipline norm, or improved for clarity.
Ok

Figures are relevant, high quality, well labelled & described.
More information should be reported on the caption of Figures 1 and 2, namely the sample size, basic statistical information, whether raw data are presented or results from your model) on Fig 1 and the R2 and p-value on Figure 2.

Raw data supplied (see PeerJ policy).
I thank you for sharing your raw data.

Experimental design

Original primary research within Scope of the journal.
Yes

Research question well defined, relevant & meaningful. It is stated how the research fills an identified knowledge gap.

Yes, although a lack of context is hampering the importance of this study. Please see my comments above.

Rigorous investigation performed to a high technical & ethical standard & Methods described with sufficient detail & information to replicate.

I have four main issues resulting in my incapacity to assess the strength and validity of the findings, mainly arising from a lack of information in the methods section. These information are also crucial to assess the repeatability of your study.

1. You did not mention how you determined age of OR kittens, for which date of birth is likely not known (you say line 62 “if known”: how did you proceed for the “not known”?). This could create a strong bias in your analyses, since date of birth is likely much more precise for MR and OR kittens. Therefore, you should address this in details in your Methods.

2. On which tissue did you measure telomere length? Did you isolate white blood cells from your blood samples? Or did you measure it on total blood? If so, please justify by witing relevant references.

3. Please provide information on whether you did the qPCR measurements in duplicates or triplicates. On one or two plates (did you use 96-plates or 384-plates)? Please also detail the composition of the Mix that you used to carry-out your qPCR, which primers and what were the cycling conditions. Please also include intra and inter-plate (if any) coefficients of variation. These are extremely important to assess the strength and the repeatability of your measurements.

4. There is no section detailing the data nor statistical analyses that you conducted. Please add one. L95-98 of the results should be included in this section but is not sufficient to assess what you did. For instance, you say that you used a generalized linear model, but I assume that you mean generalized linear mixed model, since you included a random factor? In addition, which law did you use (normal)? If so, did your residuals follow normality? I am also concerned about conducing a Linear Mixed Model on these data given the (apparent) difference in the variance and the large difference of sample size between your two groups.

General comments and questions

L70: you should indicate at which age the kittens died (to understand whether this had an influence on your experimental design and analyses or not).
L74: approximately 1 and 8 weeks > please be more specific (give the information in days±SE)

L73-82: consider providing a table of how many viable samples with extracted DNA you managed to obtain at 1 and 8 weeks, for OR and MR kittens, respectively, as it is hard to have a clear idea when reading the text.

L81-82 – that’s a lot regarding your already low sample size for MR kittens (9 at both points and 6 at one point). This is raising the question on “how many samples did you have at each point at the end?’

General question: do you have any information on whether the 42 kittens made it to adulthood? If yes, you could include an analysis with a binary response (yes/no), and looked at whether telomere length or telomere dynamics influenced survival in kittens (while comparing OR and MR kittens).

Validity of the findings

3. VALIDITY OF THE FINDINGS

Unfortunately, the lack of information on the analyses conducted, the methodology (especially on the qPCR approach), the sample size of each category does not allow me to assess the strength, validity and repeatability of the findings. I could use the supplementary datasets to conduct analyses myself but my numerous questions on qPCR analyses would in any cases do not allow me to have sufficient information to assess the validity of the findings. In addition, I think that this information should be specified in the article, so that, by reading the methods, results and looking at the figure, the reader may be able to assess whether the analyses were performed correctly. You seem to have a much greater variation on your OR groups (Fig 1) than on your MR group (Figure 1). Nonetheless, I cannot assess if this is due to a difference in sample size, an unsuitable statistical/analytical approach, to a real difference in the variance between the two groups or to measurement errors.

I appreciate your supplementary sensitivity analysis but this should be developed in the methods section and not in the results.

---

## Round 0.2 · Major Revisions

Both reviewers find great improvements in your manuscript. However, they still have some concerns, especially, in the method of statistics. Please provide the data that is statistically superior. I think that this is essential for acceptance.

Reviewer 1 ·

Basic reporting

The clarity of the MS is much-improved. I appreciated the additional background information and cited literature in the introduction. The statistical methods are much more clear and easy to assess and understand. They seem appropriate to me.

The sample sizes are much more clear, although the table does not detail the number of kittens that had samples at both time points. This info would be useful.

I do not have the expertise to assess the validity of the qPCR techniques and interpretation.

Some minor comments:

L68 "other children" should refer specifically to "children raised by their mothers".

L175 refers to an interaction but it isn't explicitly clear what the interaction is. It is detailed in the methods but should be stated again here.

L237 "influenced" is too vague - there should be specific details about such effects, including direction and magnitude if possible.

Experimental design

No further comments.

Validity of the findings

The novelty is better discussed in the context of other studies.

The conclusions are discussed in the light of limitations of the study.

There is still some weakness in contextualising the findings: I see only 4-5 citations of other studies investigating similar research questions (in the discussion). Some discussion could be provided, for example, regarding why mother loss has substantial effects in some contexts and outcomes, but this is not evident in telomere lengths.

Additional comments

Overall this is a large improvement. I see only very minor issues that could easily be resolved.

·

Basic reporting

I have reviewed the documents received. I am overall happy with the response to the reviewer comments in the letter. However, I believe that some work is still needed before the work can be published. I have done my best to provide constructive and detailed comments to the authors.

Major comments

Introduction- while I appreciate the efforts of the authors in replacing their work in a broader body of work, I am a bit disturbed by the mix of ecological literature together with medical/psychological literature. The authors could perhaps just re-organize their sections to have two different paragraphs. Or highlight that both evidence from the evolutionary ecology and psychological/medical literature demonstrate the effects of early experiences on telomere length (L55). Then the reader will be less disturbed by the integration of both studies on European starlings and children in China. I do also believe that the transition to kittens should be improved (see suggestion below).

I appreciate your conclusion, which gives a good take-home message to the reader. But your whole paper, including your discussion, should bring anyone reading the paper to the same conclusion than yours. As it is, I believe that reader would end-up being a bit confused by statistical imprecisions, differences of samples sizes (which per se can be understood and could not be a major issue, but you have to be firmer on this matter), and limits of your study as discussed in details L197-237. I provide some details to help you work on those potential issues.

Experimental design

I still have questions and concerns on the methods, especially the statistical section. This is greatly related with your goal, to compare two groups that have different experiences, but with unbalanced sample sizes. Please see my comments below.

Validity of the findings

As stated in sections 1 and 2, there is still, to my opinion, some work to be done to support the validity of your findings. Please find my detailed comments in the general comments for the author below.

Additional comments

Detailed comments:

To improve readability, you could perhaps put your degrees of freedom as indices rather than in brackets (in the abstract and the result section).
L41- leads

L44 – delete also to avoid redundancy with L45

L66 – perhaps indicate the species of horse (to be consistent with the previous section in which you provide the information)

L69-70 – perhaps before mentioning kittens you could highlight the gap in the literature > effects of maternal separation on telomere length and telomere shortening in a variety of taxa. Then you mention the goal of your study. It will smooth the transition from the state of the art, including studies on children, to your study on kittens.

L81-83 – this should go with your predictions in the following paragraph (L83-88)
Methods

L112 – a space is missing after EDTA.

L116-118 – why?

From Table 1, if you compile numbers from OR kittens at Time 1, you don’t find the n=42 mentioned L93 of your methods (either n=40 if not counting the 4 samples used to develop the protocol or n=44). Please consider adding the number of kittens sampled (in addition to the number of samples); eg: n=19 samples from 10 kittens (1 kitten for which sample is only available at time 1)

L123 – Cawthon (not Cawthorn)

L133 – please provide the brand/company and details on the model of the machine used. Was it an automated machine?

L138-144 – this should be mentioned before description of qPCR measurements
Overall, I really appreciate the supplementary section provided for telomere measurements which seems complete/detailed enough to allow repeatability.

L161 – I don’t understand, you say that you used a GLMM but that your response variable followed a normal distribution. Do you mean that you used a LMM (Linear Mixed Model) and not a GLMM? This also assumes equality of variances, and since you have an imbalance in your sample sizes, this may be an issue. Did you ran any test to investigate this (Levene or else)? In addition, how did you confirm for normality (which test) to confirm your statement 165? You respond to part of these questions in the result section, but this should be stated here.

Results

L177-188 – did you look at the interaction between sample time and kittens group here (you mentioned it for the GLMM but what about this analysis)? It is a pre-requisit if you want to then separate the analysis by group. Also, why did you add this analysis? At this stage it is not clear. Perhaps you should indicate in the methodology what it brings to your questions/predictions.

L178-179 - This apparently stronger relation could solely be explained by a smaller sample size in the MR group. No?

Discussion

Consider replacing T/S ratio by relative telomere length to facilitate the reading.

L236-237 – in which way? By increasing the variance between individuals? Within individuals but at different time points? Yes it is a possibility, but if only stated like this, you give the impression to the reader that your results are not trustworthy. You really should develop further this idea and express whether – and if so, to which extent – this can be an issue for your conclusions.

I would expect you to discuss a bit more your results:
- To which extent your study brings information to the field; eg: you can make comparisons with studies mentioned in your introduction, such as the study of Lansade et al 2018 on horses/foals. What could be the main difference? These foals were “abruptly separated” from their mother (L65). In your case kittens are indeed hand-reared and cared by humans, which could compensate in a way > although I agree with you, literature is needed to support this claim (L202).
- Provide the reader with a stronger statement that your results (Fig 1 and 2) are not caused by selective disappearance of kittens with short telomeres (because they died), or by the fact that you don’t have the same kittens measured at the two sample times. It is kind of mentioned L224-231 but perhaps to vaguely to be clear enough for the reader.
- An extra suggestion: perhaps providing in supplementary material an analysis with only data on kittens from which you have a measurement at Time 1 and Time 2 could be helpful?

In addition, consider regrouping the “limits” of your study in a single paragraph? As it is, it gives the impression than each one of your argument is counteracted by a “limit” of your study, which makes the whole discussion not very convincing. I do appreciate the transparency and I believe that these limits must be discussed, but perhaps with more implications and in a short paragraph.

---

## Round 0.3 · Minor Revisions

Reviewers find a great improvement of the manuscript. However, they still find problems with the assessment of the qPCR validity and others. I think the reviewer's comments reasonable and will improve your manuscript. Please correct your manuscript accordingly to the comments.

Reviewer 3 ·

Basic reporting

Your revised manuscript entitled “Early maternal separation is not associated with changes in telomere length in domestic kittens (Felis catus). “ (Manuscript ID 52366) deals with the effect of one stressor (early maternal separation) on one ageing parameter (telomere length TL) in kittens.

General comments: This is an interesting study, however, it has limited data and narrow approach towards studying the telomere length that may be associated with aging. They conclude that there is no effect of early maternal separation on telomere length in kittens and did not had influence between week one and week eight.

Overall the revision brings a large improvement. I see only one big issue missing to assess the validity of the qPCR (see my comments in the methods section) and some minor issues that could be easily resolved.

Introduction
With the changes made the introduction provides an adequate background information.

Methods
I have several major comments regarding the technical part: see experimental design section

Discussion
Line 251
Not well formulated. We cannot be certain that the rate of attrition in this species of bird and in the cat is comparable, in particular because the metabolism is not comparable. Furthermore, telomeres are not measured in the same cells (Lymphocyte/erythrocyte) and the rate of renewal of these cells may not be comparable.

Figures
Figure 1: Please add the mean for each category on the figure.
Figure 2: Please use the same notation as in the text, OR and MR instead of orphan Y/N.

All other comments have been addressed.

Experimental design

I still have questions and concerns on the quantification of telomere length. Please see my comments below.

Methods
I have several major comments regarding the technical part:

Line 130 – viable DNA
What do you exactly mean? Is it not rather valid DNA or to be precise good quality DNA. Could you please rewrite these sentence? How do you control the DNA quality?

Line 135 – a column that retained DNA
Could you please just add the name of the manufacturer, please?

Line 138 and line 157
You write that you dilute your DNA at 10 ng/µL in your study. Then, with the help of a liquid handler you dispatch this DNA (10 ng/µL) in a 384 plates in order to do the amplification. And line 157 you write that the concentration of DNA in all experimental reactions was 2.5 ng/µL. So did you do a dilution in between?

Line 139 to 141
What is T and S? You have to explain the abbreviation the first time you used it, please.

Line 144
Please rewrite the sentence.
“Telomere lengths were assessed by qPCR (quantitative polymerase chain reaction) and expressed as a ration (T/S) of telomere repeat copy number PCR (T) relative to a single copy gene PCR (S).” Could you also just add the formula of the calculation or at least add references about how to calculate the T/S ratio?
May be it would be more judicious to place this sentence further upstream of the chapter on the quantification of telomere length? Perhaps just after the extraction….

Line 148
What are the concentration of both primers for the single copy gene, 1mM? Did you control if the chosen “single copy gene” is non variable among your population? Did you control the size of the amplicon?

Line 155 and 156
Can you please delete it to avoid redundancy with line 148 to line 150.

Line 164
You did a standard curve. Could you please add the starting concentration of the curve and the efficiencies of the telomere and single copy gene reactions, and also the r2

Line 166
What do you mean by positive controls? What are the function of them? Is it the same as the golden sample and what are the use of a golden sample?

Line 169 – “the ratios from all sets were averaged”
I don’t understand your sentence……could you please explain.
The inter-plate CV was 3%. For what: telomeres? Single copy genes? Did you also calculate intra-plate variations? Or ICC (intraclass correlation coefficient)?

General remarks on the methods:
Can you specify the number of litters in each category in the material and method or in Table 1 and the average size of these litters (and the standard deviation). If all 6 kittens in the MR group come from the same litter this may explain the lower variance observed despite the smaller litter size.
Why there is such a large variation of age at T2? Are there differences in average age between OR and MR kittens. Please provide these information in the text or in table 1.
Rather than a regression between T1 and T2 why not do the telomere attrition between these two times and test the breeding effect of this attrition?

Validity of the findings

The conclusions are discussed in the light of limitations of the study

Additional comments

Overall the revision brings a large improvement. I see only one big issue missing to assess the validity of the qPCR (see my comments in the methods section) and some minor issues that could be easily resolved.

---

## Round 0.4 · Minor Revisions

The reviewers satisfied most revisions. However, more revisions are required by one of the reviewers. I think the point raised by this reviewer is important for the conclusion. Please revise the manuscript according to this reviewer. I encourage you to resubmit this paper.

Reviewer 3 ·

Basic reporting

All the comments have been addressed.

Experimental design

I have a question regarding the primers you used and the reply you did concerning the single copy gene primers.

Telc and Telg are telomeric primers normally used for monochrome multiplex qpcr (telomere and single copy gene assay are done in the same well, see Cawthon 2009). Also the sequences for the single copy gene given in your reply (feline GAPDH forward primer 775f and feline GAPDH reverse primer 859r) have a GC clamp. These clamps are used to artificially increase the Tm of the primers so that the multiplexing is possible (normally in multiplexing around 88°C = The 74°C acquisition signal or reads provided the Ct values for the amplification of the telomere template (in early cycles when the scg signal is still at baseline); the 88°C reads provided the Ct values for the amplification of the scg template (at this temperature there is no signal from the telomere PCR product, because it is fully melted). So the primers sequences you give in your reply are in favor of doing a multiplexing but in your paper you didn’t mention the GC clamps for your single copy primers. So did your single copy gene primers have a GC clamp or not? In your paper you didn’t do multiplex, why ?

Validity of the findings

All comments have been addressed

Additional comments

I really appreciate all the detailed answers given by the authors in order to assess the validity of the qPCR method. I am overall happy with all the response to the reviewer comments in the letter.

I have just one comment on the assessment of the DNA quality for the author’s future publications.
To do a telomere length analysis it is crucial to have good gDNA quality in term of quantity, lack of contaminants and integrity. Most of the time, especially if it’s the first time you used an extraction technique or if it’s the first time you extract gDNA from a new species, it is not sufficient to measure the gDNA quantity by a spectrophotometer like a Nanodrop and to have a look at the 260 nm/280 nm ratio and the 260 nm /230 nm ratio for assessing the gDNA to be of good quality. The ratios will inform you about the absence of contaminants but are of no clue about the integrity of the DNA. In order to check the integrity you need to do a gel (see figure 1 : DNA integrity gel in Seeker et al, 2016, Method Specific Calibration Corrects for DNA Extraction Method Effects on Relative Telomere Length Measurements by Quantitative PCR). DNA quality can be assessed using agarose gel-based separation confirming the absence of any degraded DNA in the form of a smear on the gel.

There are also some typo errors:
In line 148 : it’s 260/280 nm ratio and not 260/280 nM
In line 149 : it’s 260/230 nm ratio and not 260/230 nM
In line 161 : it’s 1µM and not 1mm

All other comments have been addressed.

---

## Round 0.5 · accepted · Accept

One of the reviewers points out typos. Please upload the corrected manuscript or correct it at the proofreading stage.

Reviewer 3 ·

Basic reporting

All comments have been addressed.

Experimental design

All comments have been addressed.

Validity of the findings

All comments have been addressed.

Additional comments

Your revised manuscript entitled “Early maternal separation is not associated with changes in telomere length in domestic kittens (Felis catus). “ (Manuscript ID 52366 4th revision) is now much clearer and better streamlined than the previous versions and it is obvious that the authors improve the manuscript. The authors have done a good job and properly addressed all my previous comments, so I have no more comments to add for the manuscript.


Sorry not to have emphasized these old typos before (pdf version):
1 Could you please be consistent throughout the manuscript in your notations? For example could you either write 0.98 or .98 as line 228 (0.98 and .97 for Time 1; 0.98 and 0.93 in Time 2). Please check throughout your manuscript (line 150, 181, 184, 228 etc… decimal numbers without zero).
2 Paragraphe 172-175: annotation for degree celsius is °C, check line 174, C is missing and in line 175 there is an inversion between the 2 letters 95C°.
3 In figure 2, scatter plot of correlations between telomere lengths at two time points.
Line 3, 28 orphaned kittens (r = 450 0.28, p = 0.14), remove 450.